# Attention-Gated Brain Propagation: How the brain can implement reward-based error backpropagation

**Isabella Pozzi**
Machine Learning Group
Centrum Wiskunde & Informatica
Amsterdam, The Netherlands
isabella.pozzi@cwi.nl

**Sander M. Bohté**
Machine Learning Group
Centrum Wiskunde & Informatica
Amsterdam, The Netherlands
s.m.bohte@cwi.nl

**Pieter R. Roelfsema**
Vision & Cognition Group
Netherlands Institute for Neuroscience
Amsterdam, The Netherlands
p.roelfsema@nin.knaw.nl

## Abstract

Much recent work has focused on biologically plausible variants of supervised learning algorithms. However, there is no teacher in the motor cortex that instructs the motor neurons and learning in the brain depends on reward and punishment. We demonstrate a biologically plausible reinforcement learning scheme for deep networks with an arbitrary number of layers. The network chooses an action by selecting a unit in the output layer and uses feedback connections to assign credit to the units in successively lower layers that are responsible for this action. After the choice, the network receives reinforcement and there is no teacher correcting the errors. We show how the new learning scheme – Attention-Gated Brain Propagation (BrainProp) – is mathematically equivalent to error backpropagation, for one output unit at a time. We demonstrate successful learning of deep fully connected, convolutional and locally connected networks on classical and hard image-classification benchmarks; MNIST, CIFAR10, CIFAR100 and Tiny ImageNet. BrainProp achieves an accuracy that is equivalent to that of standard error-backpropagation, and better than state-of-the-art biologically inspired learning schemes. Additionally, the trial-and-error nature of learning is associated with limited additional training time so that BrainProp is a factor of 1-3.5 times slower. Our results thereby provide new insights into how deep learning may be implemented in the brain.

## 1 Introduction

Artificial neural networks with many layers of neural units now attain human-level performance in speech and image recognition [1, 2, 3] and in complex computer games like Chess, Go and Starcraft [4, 5, 6]. Central to this success is the application of the error-backpropagation algorithm (EBP) [7] to efficiently assign credit to individual weights between the layers in deep networks. EBP computes the error gradient for connections between all the layers. Can the brain, with its hierarchically arranged cortical areas, solve the credit assignment problem in a similar manner?

Recent studies proposed versions of EBP for the brain that required a teacher, studying the feasibility of asymmetric feedback networks [8, 9, 10, 11] and developing variants of equilibrium propagation [12, 13, 14, 15, 16], reviewed in [17, 18]. In equilibrium propagation [12, 18], activity is first

propagated to the output layer, after which the output units are nudged in the direction of the desired target values. Feedback connections then propagate the perturbed signals back to lower layers, which can then compare activity before and after teacher intervention to estimate the error gradient for all connections. The present work addresses a number of limitations of these previous schemes. First, it avoids a teacher and replaces it by trial-and-error learning in a reinforcement learning context. Second, the scheme takes the neuromodulatory systems in the brain, such as dopamine, into account that provide a reward prediction error signal (RPE). The RPE is positive if an action selected by the network is associated with more reward than expected or if the prospects of receiving reward increase, and the RPE is negative if the outcome of the action is disappointing. These neuromodulatory signals are accessible for all neurons in the brain [19, 20, 21, 22, 23, 24, 25]. Third, it uses feedback connections to propagate attentional "credit assignment" signals from higher to lower network levels which are known to gate plasticity [26, 27, 28, 29]. When a network chooses an action, the feedback signal is strongest for neurons and synapses that are responsible for the selected action. We show how this combination of the attentional feedback signals and the neuromodulatory influence can be combined at the level of a synapse to provide a learning scheme that is equivalent to EBP, for one output unit at a time. The result is a biologically plausible synaptic update rule. The learning scheme requires weight symmetry, which can occur as the result of the BrainProp learning scheme itself. Furthermore, a previous study [30] demonstrated that the learning scheme can also function in spite of delays between the signals propagated by feedforward and feedback connections.

BrainProp builds on previous learning rules like AGREL [31] and AUGMENT [32], which also combined RPEs with attentional feedback for networks of only three layers. They solved cognitive tasks including delayed reward tasks relying on the formation of new working memories. BrainProp extends AGREL and AuGMEnT to much deeper networks (here up to 8 hidden layers). We test BrainProp on MNIST, which is a standard classification task, and also on CIFAR10, CIFAR100 and Tiny ImageNet, which are more challenging. The network is trained by providing a reward if the network chooses the correct action and withholding it upon errors. On error trials, the correct class is not revealed to the network so that less information is available than in EBP. Nevertheless, we find that BrainProp is only a factor of 1-3.5 times slower across tasks. Remarkably, BrainProp's high degree of biological plausibility is associated with a performance boost compared to the other contemporary biologically inspired learning rules. BrainProp outperforms these schemes and brings complex tasks such as Tiny ImageNet into the realm of biologically plausible learning.

## 2 From Error Backpropagation to BrainProp

Starting from EBP, we first review the weight update equations under a single-action-at-a-time reinforcement learning paradigm, and we then explain how it can map onto learning in the brain by proposing specific roles for feedback connections and neuromodulatory systems.

We start with a description of the feedforward sweep in a network with $N$ layers, where a pattern is presented to the input layer and activity then propagates towards the output layer of the network. Given a deep neural network with an arbitrary number of layers, each unit $j$ in any layer $l$ computes:

$$y_i^l = f^l(a_i^l) = f^l \left( \sum_j w_{i,j}^{l-1} y_j^{l-1} \right) , \tag{1}$$

where $f^l$ is the activation function of layer $l$, $w_{i,j}^{l-1}$ the synapse connecting the $j$-th unit in layer $l-1$ to the $i$-th unit of layer $l$, $y_j^{l-1}$ the output of the $j$-th unit in layer $l-1$. The activity vector in the output layer is used to compute the error, which represents the distance between a target vector $\mathbf{t}$ and the activity in the output layer $\mathbf{y}^N$. For networks with linear output units, which will be studied here, the sum-squared-error loss function is convenient:

$$E(w) = \frac{1}{2} \sum_n \left( t_n - y_n^N \right)^2 . \tag{2}$$

The update of the weight $w_{i,j}^{l-1}$ in Eq. 1 follows gradient descent. It is straightforward to calculate the weight update of the connections between layer $N-1$ and $N$:

$$\frac{\partial E}{\partial w_{n,m}^{N-1}} =: \Delta w_{n,m}^{N-1} = - \left( t_n - y_n^N \right) y_m^{N-1} , \tag{3}$$

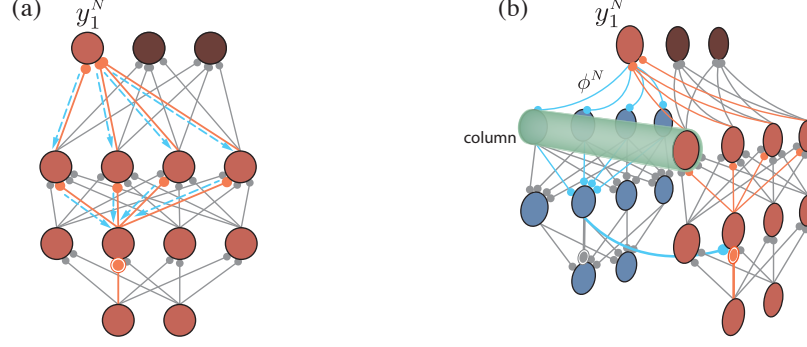

**Figure 1:** (a) EBP traversing the network from output to input; (b) backpropagation through a feedback network.

the local error $\delta_n$ of output unit $n$ is then $-\left(t_n - y_n^N\right)$ – another choice often used for EBP is the softmax activation combined with the cross-entropy loss function, which gives rise to an equivalent local error. In EBP the local error signals are propagated back from the higher layers back to the lower layers of the network and used to update all the synapses (**Fig. 1a**). The local error of a unit $i$ in layer $l$ depends on the local error signal of the units $k$ in the layer $l + 1$ above, the derivative of their activation functions, $g_k^l := f^{l'}(a_k^l)$, and the strength of the connections to these units:

$$\delta_i^l = \sum_k w_{k,i}^l \left(\delta_k^{l+1} g_k^{l+1}\right). \tag{4}$$

The weight update of all synapses follows gradient descent and it can be expressed as the product of the local error signal $\delta_i^l$ and the activity $y_j^{l-1}$ of neuron $j$ in the lower layer [33]:

$$\Delta w_{i,j}^{l-1} = \delta_i^l g_i^l y_j^{l-1}. \tag{5}$$

Supervised learning requires a teacher who reveals the correct class. We will use a simple example with only three classes to compare EBP to BrainProp. Suppose for example that the network makes an error by choosing class 1 for a particular stimulus whereas the correct choice is class 2:

$$\mathbf{t} = \begin{bmatrix} 0 \\ 1 \\ 0 \end{bmatrix}, \quad \text{and} \quad \boldsymbol{\delta}^N = \mathbf{t} - \mathbf{y}^N = \begin{bmatrix} -y_1^N \\ 1 - y_2^N \\ -y_3^N \end{bmatrix}, \tag{6}$$

and the chosen class is the one with the highest activity so that $y_1^N > y_2^N, y_3^N$. EBP will change the connections to push the activity of the first and third output unit towards zero and to increase the activity of the second unit towards one.

BrainProp is a reinforcement learning scheme and it samples a class $s$ using a stochastic action selection process in the output layer. If $s$ is correct, the network receives a reward, $r$, of 1 and rewards are withheld upon errors, but there is no teacher that informs the network about the correct choice:

$$r = \begin{cases} 1 & \text{if the correct class is selected,} \\ 0 & \text{if a wrong class is selected.} \end{cases} \tag{7}$$

Furthermore, the activity $y_s^N$ of unit $s$ in the output layer is interpreted as the current estimate of the expected amount of reward if the network selects class $s$ for the current input pattern. The difference between the amount of reward obtained and expected is the reward prediction error (RPE) and this difference will be minimized during learning:

$$E(w) = \frac{1}{2}\left(r - y_s^N\right)^2. \tag{8}$$

Suppose that the network chooses class 1. The reward expectancy is $y_1^N$ and the RPE is $-y_1^N$ because no reward is obtained. If class 2 is selected a reward of 1 is given and the RPE equals $1 - y_2^N$. The local error signals are only computed for the selected class $s$:

$$\boldsymbol{\delta}^N = (-y_1^N) \cdot \begin{bmatrix} 1 \\ 0 \\ 0 \end{bmatrix} \text{ if } s = 1; \quad \boldsymbol{\delta}^N = (1 - y_2^N) \cdot \begin{bmatrix} 0 \\ 1 \\ 0 \end{bmatrix} \text{ if } s = 2; \quad \boldsymbol{\delta}^N = (-y_3^N) \cdot \begin{bmatrix} 0 \\ 0 \\ 1 \end{bmatrix} \text{ if } s = 3. \tag{9}$$

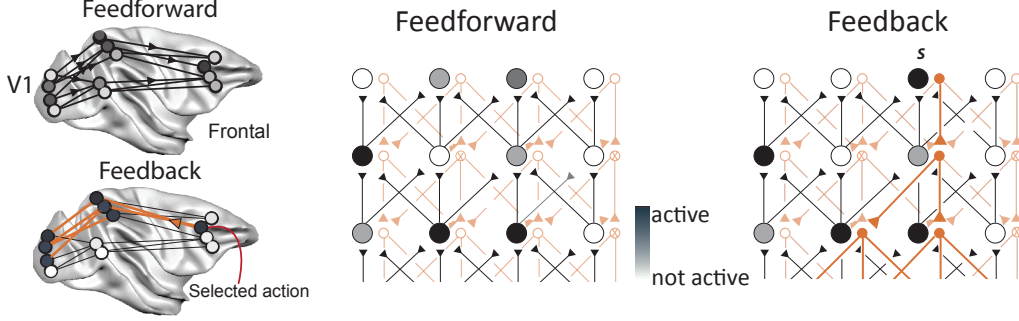

**Figure 2:** Details of the BrainProp algorithm: only one output unit, corresponding to the selected action or class, is trained at a time by means of an attentional feedback signal.

Only synaptic weights that pertain to the activity of $s$ will be updated (**Fig. 2**). In what follows we will write $\delta$ for $\delta_s^N$, because it is a scalar that represents the RPE. We know from neurophysiological work that there are circuits in the brain that code for the RPE so that Eq. 9 is in accordance with reinforcement learning [20]. Importantly, neuromodulators such as dopamine are released diffusely into the neural tissue so that they can inform all the synapses about the RPE.

We will now first determine the weight update for connections $w^{N-1}$ between layers $N-1$ and $N$:

$$\Delta w_{n,m}^{N-1} = \begin{cases} \delta \ y_m^{N-1} & \text{if } n = s, \\ 0 & \text{otherwise.} \end{cases} \tag{10}$$

If activity in the output layer becomes a one-hot vector for the selected class upon action selection, the weight update corresponds to Hebbian learning, modulated by the RPE [31, 34]. A comparison of Eq. 6 and Eq. 9 reveals that BrainProp's weight update for these connections is identical to that of EBP if all outputs are selected once and synaptic updates occur afterwards.

In general, Hebbian learning gated by the RPE has been called perturbation learning [35] and is known to perform poorly for deeper networks and for networks with many units, because synapses not involved in action selection are also modified so that the correlation between the actual and desired weight updates is diluted [31, 18]. BrainProp greatly improves the variance of the weight updates by using feedback connections from the selected actions to lower processing levels, enabling plasticity of relevant synapses only. As we will outline below, the feedback connections of BrainProp propagate activity and not error signals, consistent with neuroscientific observations. In neurophysiology, feedback influences are usually characterized as effects of top-down attention. If an action is chosen in the brain, neurons in sensory areas that provided input to this action receive an attentional feedback signal from the motor selection stage that increases their activity level (i.e. [36, 27]). The hypothesis that feedback connections gate learning is supported by psychological literature demonstrating that attention gates learning as well as the effects of feedback on synaptic plasticity [37].

We will now compute the weight update of BrainProp for the connections between layers $N-2$ and $N-1$, and then provide a recursive formula that governs the weight updates in all lower layers. In what follows we will assume that the strength of the feedback connection $w_{j,i}^{\text{FB}}$ between unit $i$ in layer $l$ and unit $j$ in layer $l-1$ is identical to the strength of the feedforward connection $w_{i,j}^{\text{FF}}$. This reciprocity of feedforward and feedback connections does not hold at the level of single neurons in the brain, but we hypothesize that it is true at the level of cortical columns. Hence, a unit of BrainProp corresponds to a cortical column in the brain with both feedforward and feedback units (**Fig. 1b**). In previous work, it was demonstrated that symmetrical weights emerge during learning using the proposed learning scheme [31] and recent studies have suggested specialized learning rules to refine this reciprocity [9]. In what follows we will use the notation $w_{i,j}$ both for $w_{i,j}^{\text{FF}}$ and $w_{j,i}^{\text{FB}}$, in particular if there is no ambiguity. We define the amount of feedback activity at the level of the output layer as:

$$\phi_n^N = \begin{cases} 1 & \text{if } n = s, \\ 0 & \text{otherwise,} \end{cases} \tag{11}$$

where unit $s$ has activity 1 upon action selection and the other output units are silent (see **Fig. 2**). Then, we can compute the amount of feedback that arrives from the selected action $s$ at unit $m$ in the

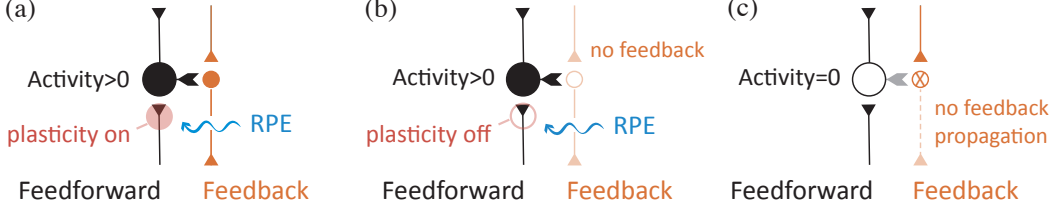

**Figure 3:** Gating mechanisms in BrainProp. (a) The activity of the feedforward neuron is above 0 and the feedback neuron propagates activity and plasticity is on; (b) if the activity of the feedforward neuron is above 0 but the feedback neuron does not receive activity, plasticity is off; (c) if the activity of the feedforward neuron is not above 0, no feedback activity is propagated.

penultimate layer $N - 1$ as:

$$\phi_m^{N-1} = \sum_n w_{n,m}^{FB, N-1} \phi_n^N = w_{s,m}^{FB, N-1} , \qquad (12)$$

where the derivative of the activation is not included as output neurons are linear. The weight update for the connection between unit $p$ in layer $N - 2$ and unit $m$ in layer $N - 1$ is gated both by $\phi_m^{N-1}$ and the derivative $g_m^{N-1}$ of the activation function of unit $m$ and equals:

$$\Delta w_{m,p}^{N-2} = \delta \, \phi_m^{N-1} \, g_m^{N-1} \, y_p^{N-2} . \qquad (13)$$

This a form of Hebbian learning, which is modulated by the RPE $\delta$ and the amount of feedback $\phi_m^{N-1}$ from the higher layer arriving at unit $m$. We will come back to the role of $g_m$ at the end of this section, and will explain how it conforms to what we know about the effect of top-down attention. The same learning rule is used for all other layers below. The amount of attentional feedback arriving at unit $i$ in layer $l$ from units $k$ in layer $l + 1$ equals:

$$\phi_i^l = \sum_k w_{k,i}^{FB,l} \, \phi_k^{l+1} \, g_k^{l+1} , \qquad (14)$$

which only depends on feedback signals from the next higher layer and the derivatives in that layer, in contrast with Eq. 4, where error signals and not activity is propagated. The weight update of a connection between a unit $i$ in layer $l$ and a unit $j$ in layer $l - 1$ is given by:

$$\Delta w_{i,j}^{l-1} = \delta \, \phi_i^l \, g_i^l \, y_j^{l-1} \; = \text{RPE} \cdot \text{feedback}_i^l \cdot \text{derivative}_i^l \cdot \text{feedforward activity}_j^{l-1} , \qquad (15)$$

where the error vector $\delta_i^l$ needed in Eq. 5 is now split into the broadcasted scalar RPE and the net feedback activity $\phi_i^l$.

We note again that this represents a form of Hebbian learning, in which plasticity is modulated by the global RPE and feedback received from the next higher layer. The weight updates of all synapses in BrainProp are identical to those of EBP, if EBP would only take the local error $\delta_s$ of the selected action into account and ignore other local errors $\delta_n$ for $n \neq s$ in the output layer. Furthermore, the weight updates of BrainProp would be identical to those of EBP if all output units are selected exactly once and weight updates applied thereafter. Thus, if BrainProp would use a suboptimal action sampling policy by first trying all $C$ classes once before updating the weights, the weight updates would be identical to those of EBP although it would require $C$ times the number of stimulus presentations. However, we will here use an action sampling strategy in which output units that receive much input (i.e. with higher activations) are more likely selected: in 98% of the cases the output neuron with the highest activity is selected and for the remaining 2% the network selects an output unit using a Boltzmann distribution over the output activations, i.e. a Max-Boltzmann controller [38]. Our simulations below show how BrainProp initially goes through a phase in which it needs to find the categories of stimuli by trial and error. Once classification starts to work, however, the learning focuses on those stimuli that are still erroneously classified and that are therefore presumably close to the category boundaries. In our experiments, the convergence rate of BrainProp turned out to be only a factor of 1-3.5 slower than that of EBP, even if there were tens to hundreds of classes.

We will now discuss the biological plausibility of $g_i$ in Eq. 15, which is the derivative of the activation function in the feedforward pathway. The factor $g_i$ influences plasticity of the column $i$ as well as the

**Table 1:** Results on fully connected (`dense`), locally connected (`loccon`) and convolutional (`conv`) networks (averaged over 10 different seeds, the mean and standard deviation are indicated).

| Dataset | Architecture | Rule | Epochs [#] | Accuracy [%] |
|---------|-------------|------|-----------|-------------|
| MNIST | `dense` | BrainProp | 51(19) | 98.68(0.07) |
| | `dense` | Error-BP | 33(19) | 98.54(0.16) |
| | `conv` | BrainProp | 63(18) | 99.31(0.04) |
| | `conv` | Error-BP | 51(13) | 99.39(0.03) |
| | `loccon` | BrainProp | 65(19) | 99.05(0.06) |
| | `loccon` | Error-BP | 70(28) | 99.06(0.06) |
| CIFAR10 | `conv` | BrainProp | 125(47) | 69.30(0.75) |
| | `conv` | Error-BP | 145(44) | 70.69(0.78) |
| | `loccon` | BrainProp | 151(35) | 64.92(0.32) |
| | `loccon` | Error-BP | 171(48) | 64.31(0.54) |
| CIFAR100 | `conv` | BrainProp | 110(34) | 34.07(0.50) |
| | `conv` | Error-BP | 102(61) | 36.56(0.87) |
| | `loccon` | BrainProp | 229(30) | 30.04(0.33) |
| | `loccon` | Error-BP | 43(26) | 31.53(0.48) |

backward propagation of activity in the feedback pathway through column $i$. Clearly, the activity level of a cortical column is a local signal, which is available to synapses and neurons in a column. However, the biological plausibility of computing the derivative will depend on the precise shape of the activation function. In simulations below, we used rectified linear units (ReLUs), which have a very simple derivative: if the feedforward units of a cortical column are not activated above their threshold, $g_i = 0$ and if they are active $g_i = 1$. In other words, plasticity is switched on for those columns that are activated by the feedforward connections (**Fig. 3a** and **Fig. 3b**), and is switched off for silent columns, in accordance with a Hebbian plasticity rule. Importantly, the inactive columns also do not propagate the feedback signal onwards to lower layers (**Fig. 3c**). This feature of BrainProp conforms with neuroscientific evidence showing that attentional feedback effects on the firing rate of sensory neurons is pronounced if the neurons are well driven by a stimulus and much weaker if they are not [39, 40, 41]. The plasticity rules for feedback connections are the same as for feedforward connections so that they remain reciprocal. It is this reciprocity that ensures that plasticity is confined to the connections between units that provided input that was pertinent to the selection of action $s$.

## 3 Experiments

We evaluated the performance of BrainProp on the MNIST, CIFAR10, CIFAR100 and Tiny ImageNet [42] data sets. The MNIST dataset consists of 60,000 training samples (i.e. images of 28 by 28 pixels), while the CIFAR datasets comprise 50,000 training samples (RGB images of 32 by 32 pixels) and Tiny ImageNet has 100,000 images (of 64 by 64 pixels) equally divided across 200 classes. Hence, the output layer had 10 units for MNIST and CIFAR10, 100 for CIFAR100 and 200 for Tiny ImageNet. For convenient implementation, we used a batch stochastic gradient decent method, but the learning scheme works equally well with learning after each trial. For each batch, we calculated the gradients, divided by the batch size, and updated the weights. We presented batches until the whole training dataset was processed, indicating the end of an epoch. At the end of each epoch, a validation accuracy was calculated on the validation dataset. We used an early stopping criterion and stopped training if the validation accuracy had not increased for 45 consecutive epochs (by the third decimal). We here used a ReLU activation function for hidden units. We used softmax output units and a cross-entropy loss for EBP and linear output units and a sum-squared-error loss for BrainProp. The code and selected pre-trained models are available at `https://github.com/isapome/BrainProp`.

In these classification tasks reward feedback is given immediately upon the choice. These tasks are simpler than more general RL settings that necessitate the learning of a number of intermediate actions before a reward can be obtained. The performance of the BrainProp-like learning rules in

**Table 2:** Results on the deeper architecture, averaged over 10 seeds, the mean(standard deviation) are indicated.

| Dataset | Rule | Epochs [#] | Accuracy [%] |
|---|---|---|---|
| CIFAR10 | BrainProp | 105(4) | 88.88(0.27) |
| | Error-BP | 97(17) | 88.48(0.55) |
| CIFAR100 | BrainProp | 218(22) | 59.58(0.46) |
| | Error-BP | 101(1) | 63.39(0.42) |
| Tiny ImageNet | BrainProp | 328(75) | 47.50(1.30) |
| | Error-BP | 101(46) | 47.35(3.06) |

shallower networks that are trained on these types of tasks has been addressed elsewhere [32]. This previous work used highly pre-processed, compact and abstracted sensory representations. The present work not only goes beyond these previous studies by generalizing the learning rule to deeper networks but also by addressing much more complex input patterns.

Since we focus on biologically plausible learning, we implemented locally connected layers for MNIST, CIFAR10 and CIFAR100 and compared performance to that of non-biological convolutional networks for which synaptic weights are shared between different image locations. For these simulations, we configured three hidden layers and for MNIST we also include a 3-layer fully connected network. The locally connected networks contained one locally connected layer with 32 filters with 3x3 kernels with batch normalization, another locally connected layer with the same parameters but with a 2x2 stride and a dropout rate of 0.3 and a fully connected layer of 500 neurons with a 0.3 dropout rate. Convolutional networks were setup identically but with shared weights. Memory limitations prevented us from implementing deeper networks with locally connected layers due to the associated vast numbers of parameters. We note that the process of batch normalization is compatible with what we know about homeostatic plasticity in the brain [43]. Dropout is also biologically plausible: by removing random hidden units in each training run, it simulates the regularisation process carried out in the brain by the stochastic firing of neurons [44]. The fully connected network consisted of hidden layers with 1,500, 1,000 and 500 units.

For the fully connected network we used a schedule with a learning rate starting 1 which was halved every 100 epochs. The weights were randomly initialized from a normal distribution with a zero mean and 0.1 standard deviation. For all the other experiments presented in this paper we used a learning rate of 0.1 as a starting value for the schedule and a standard deviation of 0.005 for the weight initialization. To fit into GPU memory, we used a batch size of 32 for locally connected networks while for all the other experiments we used a batch size of 128.

**Table 1** compares the performance of EBP and BrainProp for the fully connected networks, the locally connected networks and corresponding convolutional networks. For all the three datasets, the accuracy of BrainProp was comparable to that of EBP (**Table 1**); however, locally connected networks reached a lower validation accuracy compared to convolutional networks, with a small gap for MNIST, and larger one for CIFAR10 and CIFAR100. The speed of convergence (Epoch [#]) for BrainProp compared to EBP was comparable, except for the locally connected CIFAR100 network.

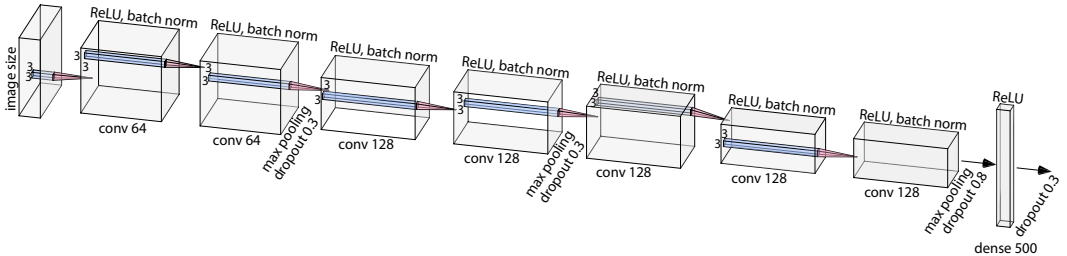

**Figure 4:** Architecture used for CIFAR10, CIFAR100 and Tiny ImageNet (the output layer, of respectively 10, 100 and 200 units, is omitted). Figure made using [45].

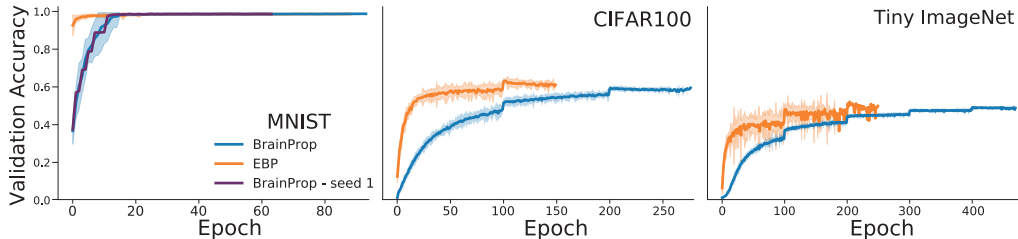

**Figure 5:** Validation accuracy trend for EBP and BrainProp. The mean ± standard deviation for 10 seeds is plotted. For `dense` MNIST, the trend obtained with a single seed is also shown.

We next ran experiments with BrainProp and EBP for CIFAR10, CIFAR100 and Tiny ImageNet with a convolutional network with eight hidden layers, arranged as shown in **Fig. 4**. We used the same hyperparameters as in the more shallow networks, but added L2 regularization of 0.0005.

As shown in **Table 2**, BrainProp reached a relatively high classification accuracy of 88.88% on the CIFAR10 task and BrainProp's speed of convergence (Epochs [#]) was a factor of 1 to 1.5x slower than that of EBP. The final accuracy obtained with BrainProp (59.58%) for CIFAR100 was slightly lower than that with EBP (63.39%) and the convergence rate that was between 2 and 2.5x slower. BrainProp's accuracy on Tiny ImageNet was 47.50%, comparable to EBP's accuracy of 47.35%, and learning was a factor of 3 to 3.5x slower.

**Fig. 5** compares BrainProp's learning process to that of EBP for the MNIST, CIFAR100 and Tiny ImageNet benchmarks (CIFAR10 not shown). The "steps" in the curve of BrainProp, which are easiest to observe for MNIST, are of interest: the network appears to "discover" the classes, one at a time. In contrast, EBP's teacher will give feedback about the correct class the first time it presented and these steps do not occur. Learning for BrainProp is slower than for EBP. This slowdown occurs because BrainProp trains connections for only one output unit at any one time. This cost is offset by the biological plausibility of BrainProp: the learning process matches what we know about how top-down connections and neuromodulators determine plasticity in the brain [27].

## 4 Discussion

An important question in computational neuroscience is if and how the human brain with its many layers between sensory input and motor uses plasticity rules that are as powerful as EBP. BrainProp differs from previous biologically inspired learning schemes, which used a teacher to determine the local errors in the output layer and to propagate these errors to lower network layers. Instead, BrainProp is a reinforcement learning scheme in which a choice is made for an action in the output layer and an attentional feedback signal determines which units at lower layers contributed to this action. As a result, only relevant connections are changed, which causes variance of the weight updates to be much smaller than schemes that only use the RPE, such as REINFORCE [46, 31]. BrainProp is thus able to train deep networks for a wide range of challenging classification problems.

In BrainProp, only units that receive the backpropagated attentional feedback signal, $\phi$, that originates from the selected action become sensitive to the RPE (i.e. the global $\delta$). In the brain, the RPE is mediated by neuromodulatory signals that are released diffusely to make them available to all synapses. These signals determine whether the relevant synapses (with high $\phi$) increase or decrease in strength. Even though BrainProp is mathematically equivalent to EBP for one output unit at a time, it is also biologically plausible. This biological plausibility derives from (1) its use of the neuromodulatory signals known to determine plasticity [27], (2) the use of attention known to gate learning, (3) the availability of all signals that determine plasticity at the synapses and (4) trial-and-error learning, so that it can work without a teacher providing a target signal to every output unit.

BrainProp requires feedforward and feedback connections that are approximately reciprocal. In previous work, it was demonstrated that such symmetrical weights can emerge during learning using the proposed learning scheme [31] and recent studies have suggested that the brain may even have specialized learning rules to calibrate this reciprocity [9]. In the brain, the approximate reciprocity of connections may hold at the level of cortical columns, but not at the level of individual neurons.

Hence, the units of BrainProp should be identified with cortical columns that consist of hundreds of cells and not with individual neurons.

BrainProp is a generalization of existing learning rules AGREL [31] and AuGMEnT [47, 32], which have been used previously to train networks with one hidden layer. The generalization to many layers, achieved by developing a framework that gates the attention signals sent back through multiple layers, greatly expanded BrainProp's capacity as a learning scheme. We showed how it trains deep networks to perform the MNIST, CIFAR10, CIFAR100 and Tiny ImageNet tasks. BrainProp attained an accuracy that is on par with EBP, even for Tiny ImageNet with its 200 classes. BrainProp thereby outperforms other biologically plausible learning schemes and it may be the first biologically plausible scheme that can train networks on larger problems, such as Tiny ImageNet [10]. A remarkable finding is that the trial-and-error nature of learning of BrainProp incurred a very limited cost of 1-3.5x more training epochs, even if there were 200 classes that had to be found by trial and error. BrainProp learns about classes that are correctly or erroneously selected and will focus learning on miss-classified stimuli. Whereas we used BrainProp to train network on classification tasks, defined as direct reward problems, future work can use versions of the algorithm to train more complex cognitive tasks in which rewards are delayed and depend on a number of consecutive actions. These explorations could be based on the AuGMEnT framework, which trains networks on delayed reward tasks and even allows networks to form working memories if needed (such as e.g. in POMDPs).

We conclude that the present and related work on biologically plausible learning is starting to bridge the gap between learning in machines and in the brain. Insights from the machine learning and neuroscience fields is contributing to a genuine understanding of learning in the brain, with its many processing stages between sensory neurons and the motor neurons that ultimately control behavior.

## Broader Impact

Our research addresses how deep learning can be implemented by the brain. It does not only address of the biggest unsolved mysteries related to how our brain, with its many layers between input and output learns, but it may ultimately also shed light on conditions in which learning is impaired. For example, our work suggests that attention is important for learning. It may thereby inspire new research into how e.g. attention deficits impair learning. Our work also suggests an important role for neuromodulatory signals, such as dopamine, in learning. Our work may provide insight into how diseases that impair the neuromodulatory systems (e.g. Parkinson's disease) cause learning deficits.

## 5 Acknowledgments

We thank Walter Senn for his constructive feedback. This work was financed by NWO NAI grant 656.000.002 and H2020 HBP SGA3 project 945539.

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
