[Supplementary Material · README.pdf]

# Training deep networks with a biologically plausible algorithm

*Implementation of BrainProp, a biologically plausible learning rule that can train deep neural networks on image-classification tasks (MNIST, CIFAR10, CIFAR100, Tiny ImageNet).*

## BrainProp: How the brain can implement reward-based error backpropagation

This repository is the official implementation of "BrainProp: How the brain can implement reward-based error backpropagation".

In the paper we show that by training only one output unit at a time we obtain a biologically plausible learning rule able to train deep neural networks on state-of-the-art machine learning classification tasks. The architectures used range from 3 to 8 hidden layers.

## Requirements

The current version of the code requires a recent (as of June 2020) version of tensorflow-gpu, CUDA and cuDNN and it was specifically tested on the following versions of the packages:

- Python 3.6.6
- pip 20.1.1
- CUDA 10.1.243
- cuDNN 7.6.5.32

To install the required libraries and modules (after having created a virtual environment with the versions of Python and pip indicated above):

```
pip install -r Requirements.txt
```

### Datasets

- MNIST, CIFAR10 and CIFAR100 are automatically available through keras.
- Tiny ImageNet can be downloaded from the official page of the challenge or extracted by running:

```
python tinyimagenet.py
```

in the directory where the file "tiny-imagenet-200.zip" is located.

## Training and Evaluation

To train the model(s) in the paper, run this command:

```
python main.py <dataset> <architecture> <algorithm>
```

the training will stop when the validation accuracy has not increased for 45 epochs, otherwise until 500 epochs are reached.

The possible `<dataset>` - `<architecture>` combinations are:

- `MNIST` -{`dense`, `loccon`, `conv`}
- `CIFAR10` -{`loccon`, `conv`, `deep`}
- `CIFAR100` -{`loccon`, `conv`, `deep`}

- `TinyImageNet` - `deep`

  For the details of the architectures, please refer to the paper.

For `<algorithm>`, set `BrainProp` for BrainProp or `EBP` for error-backpropagation.

Add the flag `-s` to save a plot of the accuracy, the trained weights (at the best validation accuracy) and the history file of the training.

To load and evaluate a saved model:

```
python main.py <dataset> <architecture> <algorithm> -l <weightfile.h5>
```

Three pre-trained models (on the deep network with BrainProp) on CIFAR10 (`CIFAR10_BrainProp_weights.h5`), CIFAR100 (`CIFAR100_BrainProp_weights.h5`) and Tiny ImageNet (`TIN_BrainProp_weights.h5`) are included.

All the hyperparameters (as specified in the paper) are automatically set depending on which architecture is chosen.

## Results

All the experiments ran on one node with a NVIDIA GeForce 1080Ti card.

Our algorithm achieved the following performances (averaged over 10 different seeds, the mean and standard deviation are indicated):

| BrainProp | Top 1 Accuracy [%] | Epochs [#] | Seconds/Epoch |
| --- | --- | --- | --- |

| BrainProp | Top 1 Accuracy [%] | Epochs [#] | Seconds/Epoch |
|---|---|---|---|
| MNIST - `conv` | 99.31(0.04) | 63(18) | 3 |
| CIFAR10 - `deep` | 88.88(0.27) | 105(4) | 8 |
| CIFAR100 - `deep` | 59.58(0.46) | 218(22) | 8 |
| Tiny ImageNet - `deep` | 47.50(1.30) | 328(75) | 47 |

For the `dense` and `conv` simulations the speed was 3s/epoch, while for `loccon` the speed ranged between 45- and 60s/epoch.

For the complete tables and figures, please refer to the paper.

| BrainProp | Top 1 Accuracy [%] | Epochs [#] | Seconds/Epoch |
|---|---|---|---|
| MNIST - `conv` | 99.31(0.04) | 63(18) | 3 |
| CIFAR10 - `deep` | 88.88(0.27) | 105(4) | 8 |
| CIFAR100 - `deep` | 59.58(0.46) | 218(22) | 8 |
| Tiny ImageNet - `deep` | 47.50(1.30) | 328(75) | 47 |