[Reviews · NeurIPS 2020]

Review 1

Summary and Contributions: The paper proposes a model of biologically plausible learning of classification and selection tasks. One feature of the model is that it does not provide a supervision signal, but only a reward-prediction error signal related to the selected class. Another feature is that it does not backpropagate errors, but rather neurons' degrees of involvement in decisions. This feedback signal is converted into an error gradient via multiplication with a global reward-prediction error. The model is tested on standard image classification tasks. It is found to converge to good performance and to learn only moderately more slowly than standard supervised learning with backpropagation.

Strengths: - It seems surprising and important that learning was not much slower when error feedback was given only for the selected class, rather than for all classes. - It is an interesting approach to decouple the strength of a neuron's involvement in a decision (which is fed back) from the reward (which is widely available).

Weaknesses: - The work assumes that the feedback weights are identical to the feedforward weights (line 119), but I believe this has been the community's main complaint about the biological plausibility of backpropagation, which other papers have tried to solve in recent years. The title of the paper doesn't really seem appropriate given this context. Specifically, it seems to me that the paper more clearly addresses reward-based classification than the backpropagation problem. Activations are backpropagated rather than errors, but if identical weights are needed I don't think this is any more biologically plausible than the usual backpropagation. - Relatedly, three arguments were given to support these tied weights, but I think they need deeper explanation. The most promising seems to be from ref. [30] -- could that explanation be expanded? I didn't follow the rationale re. columns. Feedforward weights aren't even shared within a column, so how does this help with feedback weights? Ref. [9] is relevant, but it applies to error backpropagation. If it generalizes to the present model, this could be shown explicitly. - An important result is that BrainProp is only 1.5-3.5 times slower than standard backprop, but I wasn't fully comfortable with how "slower" was defined. In Figure 5, it looks like the BrainProp curves would have to be compressed by a factor of five or ten to best fit the backprop curves. "Slower" seems to be defined in terms of the early-stopping criterion (i.e. no improvement for 45 epochs). But this criterion would be relatively more aggressive in the context of BrainProp's slower learning. It isn't straightforward to compare learning rates, but as this is a key result, I think it would make sense to report results of calculating it in a couple of different ways.

Correctness: Yes, I think so.

Clarity: Yes.

Relation to Prior Work: I think more could be said about the differences with [30] in particular.

Reproducibility: Yes

Additional Feedback: - To me, the fact that learning was not much slower than standard supervised learning seems like the most important result of the paper, and I would have liked to see more analysis of how this works (rather than just a report of the empirical result). Additionally it would be nice to see a more systematic exploration of how this scales with the number of classes, including greater numbers of classes. - Line 37: "These neuromodulatory signals are accessible for all neurons in the brain." This is an important and strong statement about physiology, but I'm not sure the references support it. Many references are given, but this isn't the main topic of any of them. I looked fairly carefully for support for this statement in the first reference and didn't find it. I think ideal support would include studies on the distributions of dopamine neuron terminals and dopamine receptors. - Please confirm that g is meant to be included in eq. (5) - A small quibble with line 82: you said earlier you were focusing on linear output, so I take the first and third units could be -ve and increase toward 0, rather than decrease. - Line 111: "In neurophysiology, feedback influences are usually characterized as effects of top-down attention." I think "usually" may be going too far. It excludes predictive coding and feedback alignment perspectives, for example. Also, some physiological models in vision treat feedback as part of a dynamical system in which related lower-level features and higher-level features reinforce each other, e.g. https://www.ncbi.nlm.nih.gov/pmc/articles/PMC6635809/ - I think the model implies that when sensory neurons contribute to a movement or decision, their activity (or maybe the activity of other neurons in the column) should be reliably enhanced a short time later. Could the authors comment on neurophysiological evidence for this? - Is there a -ve sign missing from (3)? - The paper says that the plasticity rule is Hebbian (e.g. line 144), but I think this needs more explanation. There is a product of feedforward activity and feedback, but the feedback is a distinct signal that doesn't have a monotonic relationship with the neuron's feedforward output. If I understand correctly it would have to be carried within the cell (to the plastic synapses) by some signal other than membrane potential. Is there a candidate? This picture seems even less clear in light of the suggestion that the mechanism may operate at the column rather than the cell level. I don't understand how the feedforward and feedback signals are thought to meet at the synapse. - UPDATE AFTER REBUTTAL: The authors' responses were convincing. The additional physiological discussion is appreciated, and it is a particularly nice surprise that BrainProp seems to learn approximate weight symmetry.


Review 2

Summary and Contributions: This paper introduces BrainProp as a biologically inspired approximation of backpropagation. The paper shows that reward-based learning can be implemented by modulating Hebbian learning through a global reward prediction error (RPE) which replaces the error signal in BP and an attentional signal that gates synaptic plasticity. Results show that performance on challenging tasks using deep architectures is on par with standard BP while being a factor of 3 slower.

Strengths: This paper connects biologically plausible mechanisms (RPE, attentional gating, reward-based learning) to gradient-based learning. Importantly, it shows that BrainProp continues to work well in challenging settings, where alternative approximations tend to break down. The paper builds a strong case that the used mechanisms are biologically plausible and makes an important contribution to recent work which focuses on understanding how gradient-based learning could be implemented in the brain. As such, I consider this to be of high relevance to the NeurIPS community.

Weaknesses: It would be of interest to have a more formal comparison between how the BP error signal relates to the RPE x FB signal.

Correctness: Claims and method are correct as far as I can tell. A critical note on the remaining challenges would be helpful. E.g. does it generalize to spiking neural networks, how to apply this approach in recurrent networks without the need to backpropagate through time, how do single neurons have access to the symmetric feedback weights in cortical columns, etc?

Clarity: This is a clear and well-written paper.

Relation to Prior Work: The paper build on a previous algorithm for single-layer networks which consists of similar ingredients. As far as I can tell, the present work mainly generalizes this approach to deep networks. That being said, this is of course essential for demonstrating scalability of the approach.

Reproducibility: Yes

Additional Feedback: Results on learning of symmetric feedback weights would be helpful. The authors state that REINFORCE would not work well in the present setting. This and possibly other brain-inspired alternatives to BP such as targetprop would make for interesting additional baselines.


Review 3

Summary and Contributions: This paper trains deep neural network models for image classification using BrainProp, a variant of a learning algorithm previously known as AGREL. In the models considered here, a classification decision is made stochastically; a reward is then provided if the decision is correct. Experiments reveal that this does not impair learning considerably in moderate difficulty problems. A discussion is provided on the biological plausibility of the learning algorithm. *** Update: having read the authors' response I have raised my score.

Strengths: - Reinforcement learning (as opposed to supervised learning) is a well-motivated paradigm in neuroscience. - The empirical evaluations this paper contributes are a welcome addition to the field. - The biological interpretation of the learning algorithm as feedback-gated Hebbian learning is interesting.

Weaknesses: - As detailed under "related work" below, the conceptual novelty of the paper is limited, if I understood the paper correctly. I'm happy to revise this point after reading the authors' view on it. - It is unclear how to generalize BrainProp to continuous action spaces. - The paper currently suffers from some clarity and correctness issues, detailed in the appropriate sections below.

Correctness: l. 28: I found this brief description of equilibrium propagation misleading, if not wrong. This description gives the sense that in equilibrium propagation the activity propogates first forward, and then backward with the teacher present. In general, both forward and feedback propagation occurs in both phases (with or without teacher). An additional detail: output neurons are not clamped to the target, but pushed towards it. Eq. 3: It appears that a minus sign is missing; the weight update should follow the negative of the gradient. l. 73: Why do we have a minus sign here? Note that we do not have one on \delta_n for the squared error loss. l. 83: For the linear output model studied in this section, not necessarily; it could be that y_2^N is overshooting (bigger than 1) and thus needs to be decreased as well. l. 152-155: I'm confused now, do the authors use Max-Boltzmann action selection, or do they use \epsilon-greedy (the 98% vs 2%, remark before)? Using \epsilon-greedy seems to be a small difference against the original AGREL study. - On the equivalence to backpropagation: Since y^N seems to have a double meaning (pre- vs. post-action selection, see comment above) a strict equivalence between the weight updates that arise from gradient descent on (2) using a linear output, and BrainProp (which involves an action selection step that changes y^N) seems incorrect. In other words, the equations look the same because y^N was given a double meaning, but in fact, comparison of the two \Delta W would reveal a difference. I would suggest looking at the expected weight update, having appropriately defined the stochastic action selection mechanism.

Clarity: - I missed some definitions, that are left for the reader to guess. For example, how are the various transfer functions f^l chosen, after all, when discussing BrainProp, in particular f^N? l. 101 is particularly confusing: does this mean that we redefined y^N to be a sample one-hot vector instead of how it is defined in Eq. 1? Furthermore, how exactly does action selection work? It would also be helpful to comment on what restrictions we have on the action selection process, so that BrainProp is a sound algorithm. - In my opinion it would be better (especially for the NeurIPS audience, I feel) to succintly first present the BrainProp algorithm, and then perhaps work out an example step-by-step. I admit that the example way of presenting could help some readers, but I missed a concise algorithmic description.

Relation to Prior Work: It is not clear to me why BrainProp is considered an extension of AGREL (e.g., l. 268). Is BrainProp AGREL, and is this then a study of the scalability of AGREL to deeper and more complex neural network models and datasets? If not, please clarify what are the exact differences. Otherwise, it should be stated that Sect. 2 is a recap of AGREL. I find this point somewhat troubling given that the abstract presents BrainProp "as a new learning algorithm".

Reproducibility: Yes

Additional Feedback:


Review 4

Summary and Contributions: The paper introduces BrainProp, a biologically plausible algorithm relying on rewards rather than global teaching signals. A mathematical equivalence of BrainProp and backpropagation is shown. The authors demonstrate their approach experimentally on standard classification benchmarks (MNIST, CIFAR-10, CIFAR-100, Tiny ImageNet).

Strengths: This paper addresses an interesting issue in neuroscience: the fact that conventional training in deep learning requires a global teaching signal at each training step. The proposed approach avoids a teacher and replaces it by trial-and-error learning. It shows how supervised learning can be achieved in a context of reinforcement learning using rewards for actions selected, but no global teaching signal. Experiments show that BrainProp achieves equivalent accuracy to the conventional error backpropagation (EBP) algorithm. Furthermore, they find that the convergence rate of BrainProp is only a factor of 1-3.5 slower than that of EBP, even with hundreds of classes. I found this result surprising and quite interesting. The experimental study is conducted thoroughly, and includes experiments with local convolutions, which is much more biologically realistic than using shared filters.

Weaknesses: The paper does not discuss whether the BrainProp algorithm, presented as a reinforcement learning scheme, could actually be applied to standard RL tasks (rather than classification tasks).

Correctness: The biological plausibility of the proposed algorithm is supported with physiological findings. One benefit of BrainProp put forward is that experiments achieve better accuracy than state-of-the-art biologically inspired learning schemes: "Remarkably, BrainProp’s high degree of biological plausibility is associated with a performance boost compared to the other contemporary biologically inspired learning rules. BrainProp outperforms these schemes and brings complex tasks such as Tiny ImageNet into the realm of biologically plausible learning." However it was not clear to me if BrainProp addresses any of the issues that these other biologically motivated algorithms address, e.g. the weight transport problem? In the experiments, are the forward and backward weights tied, or do you use distinct forward and backward connections? Using different feedforward and feedback connections seems essential here to compare with the feedback alignment algorithm for example. Lines 152-155: “We will here use an action sampling strategy in which output units that receive much input (i.e. with higher activations) are more likely selected: in 98% of the cases the output neuron with the highest activity is selected and for the remaining 2% the network selects an output unit using a Boltzmann distribution over the output activations, i.e. a Max-Boltzmann controller” This choice seems rather arbitrary. Is there any motivation for choosing 98% and 2% ? What happens if we were to choose different numbers?

Clarity: I found the paper very clear overall and pleasant to read.

Relation to Prior Work: The authors mention in the introduction and the conclusion that BrainProp builds on existing learning algorithms like AGREL and AuGMEnT, which were previously used to train shallow networks. It would be good to give more context and details here. How does BrainProp compare to AGREL and AuGMEnT? What is the new element (if any) in BrainProp that enables to train deep networks and not just shallow ones? Another point. Rather than opposing their work to other biologically motivated algorithms (as mentioned earlier), I think it would make more sense to emphasise how this approach can complement other approaches. I see no reason why BrainProp could not be combined with feedback alignment for example.

Reproducibility: Yes

Additional Feedback: The authors could have chosen a more specific name for their algorithm ("BrainProp" is too generic). I would consider raising my score if the authors better explain how BrainProp improves on prior works such as AGREL. Minor comments. There seems to be a mistake at lines 71 and 73. The error term with the sum-squared-error loss has a + sign whereas the error term for the softmax + cross-entropy loss has a - sign. I would expect that the sign should be the same in both cases. Am I missing something? For better readability in equations 4 and 5, I would suggest to choose the relative positions of g_k and delta_k consistently (either g_k delta_k or delta_k g_k). === Post Rebuttal === Thank you for the rebuttal, and congratulations on the nice work! I encourage you to change the name of your algorithm, and choose one that better shows the link with the AGREL algo.

[Author Response · NeurIPS 2020]

We thank the reviewers for their constructive comments. Here we will focus on the main concerns. We will address
smaller points and clarity of the equations, use of minus signs and symbols into account when we revise the paper.

**1) Weight symmetry: R1,2,4** raised questions about the neuroscientific plausibility of weight symmetry and if small
deviations from symmetry in the brain would be detrimental. BrainProp does not suffer from the weight transport
problem but solves it because synaptic updates of feedforward and feedback connections are proportional so that their
strength becomes proportional during learning. While exact reciprocity/proportionality of feedforward and feedback
connections is not present at the level of single neurons in the brain, such reciprocity at the level of cortical columns
(with many connections in both directions) is plausible. Indeed, the columns have mechanisms to switch on plasticity,
such as specific types of interneurons in a column that gate plasticity, as outlined in ref. 27. **R1** asked us to expand on
evidence for reciprocity of feedforward and feedback connections. The selective attention literature provides strong
evidence for this (e.g. ref. 27). We will here illustrate it for eye movements (our revision will mention feature-based
attention too). If an object is selected in the frontal eye fields (FEF) for an eye movement, neuronal response in early
visual cortex elicited by the same object are enhanced a short time later (as anticipated by **R1**) (ref. 29). Moore and
colleagues used microstimulation in the FEF [1] and observed that feedback is channeled to a tiny hotspot in the visual
field where activity is enhanced, implying a exquisite specificity of feedback connections. In the neuroscience field,
the consensus is that attentional feedback selectively reaches those neurons in sensory cortex that gave input to the
motor response (just as in BrainProp). Importantly, these neuroscientific findings imply reciprocity of feedforward and
feedback connections. **R1** was interested in how feedback connections, which carry a distinct signal, influence plasticity.
This is a neuroscientific finding, which has been reviewed in e.g. ref. 27, which also discusses cellular mechanisms.
**R1,4** asked for results on learning of symmetric weights. In the few days before the deadline of this rebuttal, we ran
an experiment on CIFAR10 on the smaller convolutional architecture with randomly initialized feedback weights and
weight decay (untying the weights): we reached equivalent accuracy, and conclude that BrainProp learns approximate
weight symmetry. Feedback alignment fails on simple problems and is known not work at all in deeper networks.

**2) Comparison to AGREL: R1,3,4** asked about the relation between BrainProp and AGREL. BrainProp follows from
new insights of how plasticity can be switched on/off in the appropriate columns in a deep network (Fig. 3), whereas
AGREL dealt with a single hidden layer. In spite of weight updates based on information that is locally available at the
synapse, BrainProp solves many tasks that have not been yet been learned by competing schemes.

**3) Learning speed: R1** would like to see a more thorough analysis of learning speed compared to EBP. The apparently
larger gap at the start of learning is caused by the initial slow learning phase of BrainProp, when it has to find classes by
trial and error because there is no teacher. Thereafter, it catches up by only making errors at category boundaries, i.e.
images that matter to get better. Our revision will include a thorough analysis of the initial and later learning speed and
how it depends on the number of classes (note that we have already tested problems with 10, 100 and 200 classes).

**4) Homogeneity of neuromodulatory effects: R1** asks about the homogeneity of e.g. dopamine influences. The
release of neuromodulators is indeed homogeneous, see e.g. [2] and ref. 20.

**5) Formal comparison to BP and notation of $y^N$: R2** asked us to clarify the formal comparison between RPE x FB
and EBP. We will improve our description. If we sample all actions uniformly and synaptic updates occur afterwards,
the result is mathematical equivalence to EBP. We agree with the point of **R3** about the notation of $y^N$ and will improve
it by introducing an extra symbol for the one-hot post selection activity vector.

**6) Action selection: R3,4** ask us to clarify the action-selection process. We included 2% of explorative choices, for
which the network used a softmax function (mostly sampling nearby categories) and the other 98% of choices were
greedy. The results do not depend critically on this percentage of explorative choices. We will include a succinct
algorithmic description as suggested by **R3**.

**7) Equilibrium propagation**: We agree with **R3** and will improve our description.

**8) Classical RL tasks: R4** asks about the application of BrainProp to classical RL tasks. This was the topic of
AuGMEnT, a learning scheme that works for a network with one hidden layer. When comparing BrainProp to
AuGMEnT it follows that BrainProp should be able to solve challenging RL tasks, although we have not yet explored
this. Future work might consider e.g. Atari games because BrainProp can train networks with sufficient depth to build
useful internal representations of task events.

**9) Future challenges: R2** suggests to include a discussion of future challenges, such as generalization to spiking
neurons. We have initial results on this, where excitatory feedback neurons carry a linear signal of limited dynamic
range by using a fixed activity offset, similar to baseline activity of dopamine neurons (ref. 19). Moreover, Brainprop
generalizes to continuous actions spaces (point raised by **R3**) when applied to the actor-critic RL paradigm Deep
Deterministic Policy Gradient (DDPG). BrainProp could be used for a biologically realistic implementation of DDPG
in deeper networks.

[1] Armstrong, Fitzgerald and Moore, *Changes in visual receptive fields with microstimulation of frontal cortex.*
[2] Gaspar, Stepniewska and Kaas, *Topography and collateralization of the dopaminergic projections to motor and lateral prefrontal cortex in owl monkeys.*


[Meta-Review · NeurIPS 2020]

The reviewers agreed that this paper provides an important contribution to the biological learning literature, and agreed that it should be accepted. However, the reviewers were also in agreement that the authors must do the following for the camera-ready version of the paper: 1) Provide greater clarity that this is an extension of AGREL and does not involve any changes to the core AGREL algorithm, but rather, a means of gating the attention signals sent back through multiple layers. which helps for deeper networks. 2) Be clear that this is a model for deep RL, but only for one-hot policies. 3) Be clear that this is not a complete solution to the question of deep credit assignment and doesn't address some outstanding questions, most notably, the question of symmetric weights and the timing of feedback vs. feedforward signals. 3) The BrainProp name needs to change - it is too generic a name and tells readers very little about the algorithm.